# Imbalanced IL-1B and IL-18 Expression in Sézary Syndrome

**DOI:** 10.3390/ijms24054674

**Published:** 2023-02-28

**Authors:** Kelly Cristina Gomes Manfrere, Marina Passos Torrealba, Frederico Moraes Ferreira, Emanuella Sarmento Alho de Sousa, Denis Miyashiro, Franciane Mouradian Emidio Teixeira, Ricardo Wesley Alberca Custódio, Helder I. Nakaya, Yasmin Alefe Leuzzi Ramos, Mirian Nacagami Sotto, Anders Woetmann, Niels Ødum, Alberto José da Silva Duarte, José Antonio Sanches, Maria Notomi Sato

**Affiliations:** 1Laboratory of Medical Investigation LIM 56, Tropical Medicine Institute of São Paulo, University of São Paulo Medical School, São Paulo 05403-000, Brazil; 2Laboratory of Medical Investigation LIM 50, University of São Paulo Medical School, São Paulo 05403-000, Brazil; 3Cutaneous Lymphoma Clinic, Hospital das Clínicas, Department of Dermatology, University of São Paulo Medical School, São Paulo 05403-000, Brazil; 4Department of Immunology, Institute of Biomedical Sciences, University of São Paulo, São Paulo 05403-000, Brazil; 5Hospital Israelita Albert Einstein, São Paulo 05652-900, Brazil; 6Department of Pathology, University of São Paulo Medical School, São Paulo 05403-000, Brazil; 7LEO Foundation Skin Immunology Research Center, Department of Immunology and Microbiology, University of Copenhagen, 2177 Copenhagen, Denmark

**Keywords:** Sézary syndrome, lymph nodes, inflammasome, IL-1B, IL-18, erythroderma skin

## Abstract

Sézary syndrome (SS) is a rare and aggressive type of cutaneous T-cell lymphoma, with an abnormal inflammatory response in affected skin. The cytokines IL-1B and IL-18, as key signaling molecules in the immune system, are produced in an inactive form and cleave to the active form by inflammasomes. In this study, we assessed the skin, serum, peripheral mononuclear blood cell (PBMC) and lymph-node samples of SS patients and control groups (healthy donors (HDs) and idiopathic erythroderma (IE) nodes) to investigate the inflammatory markers IL-1B and IL-18 at the protein and transcript expression levels, as potential markers of inflammasome activation. Our findings showed increased IL-1B and decreased IL-18 protein expression in the epidermis of SS patients; however, in the dermis layer, we detected increased IL-18 protein expression. In the lymph nodes of SS patients at advanced stages of the disease (N2/N3), we also detected an enhancement of IL-18 and a downregulation of IL-1B at the protein level. Moreover, the transcriptomic analysis of the SS and IE nodes confirmed the decreased expression of *IL1B* and *NLRP3*, whereas the pathway analysis indicated a further downregulation of *IL1B*-associated genes. Overall, the present findings showed compartmentalized expressions of IL-1B and IL-18 and provided the first evidence of their imbalance in patients with Sézary syndrome.

## 1. Introduction

Sézary syndrome (SS) is a rare and aggressive type of cutaneous T-cell lymphoma (CTCL) characterized by an intense pruritus, erythroderma (>80% involvement of the body surface), and the systemic dissemination of clonal CD4+ T cells into the blood and, often, the lymph nodes. The median survival for SS patients is approximately 1–5 years [1], with bacterial sepsis being the main cause of death [2].

The impairment of cellular immunity has long been described in SS [3] and contributes to the significant morbidity and mortality associated with infectious complications in SS patients [4]. Dysfunctional levels of innate and adaptive circulating leukocytes, including altered Toll-like receptor (TLR) activation responses [5] with a significant reduction in absolute cell counts, B cells and impaired functional activity of natural killer cells [6,7], were found in patients with SS. Together with the tumor microenvironment, the cytokine signaling pathways may promote an inflammatory response via T-cell proliferation, leading to clonal malignant T cells with continuous expansion [8]. The phenotypic plasticity of Sézary cells has been observed in SS patients as corresponding to classical and nonclassical T-helper subsets at different molecular levels and with varied maturation phenotypes [9].

Inflammasomes are multimeric cytosolic protein complexes that assemble in response to damage-associated molecular patterns (DAMPs) and pathogen-associated molecular patterns (PAMPs), leading to the activation of inflammatory responses. Inflammasome complexes are assembled upon the activation of certain nucleotide-binding domains, leucine-rich repeat-containing proteins (NLRs), AIM2-like receptors (ALRs), or pyrin. Inflammasomes include NLRP1, NLRP3, NAIP/NLRC4, NLRP6, NLRP12, AIM2 and the PYRIN inflammasome. Specific members of the IL-1 family are expressed by cells as cytosolic pro-forms that require cleavage for the secretion of their active forms. This assembly leads to the activation of caspase-1, which promotes the maturation and release of the inflammatory cytokines IL-1B and IL-18, as well as the inflammatory cell death [10]. IL-1B is primarily produced by myeloid cells [11,12], while IL-18 is constitutively expressed in myeloid cells and epithelial cells, such as keratinocytes. IL-18, a member of the IL-1 superfamily, is a pro-inflammatory cytokine that is stored in the cytosol of producing cells, similar to IL-1B, but as opposed to IL-1α or IL-33, IL-18 is produced as a biologically inactive precursor [13,14]. IL-18 is able to cooperate with IL-12 to induce IFN-γ production, leading to NK-cell activation, T-helper type-1 cell skewing, upregulated antigen presentation, and antiviral and antitumor functions [15]. In the absence of IL-12, IL-18 can also stimulate Th2 immune responses. The activity of IL-18 is regulated by the endogenous inhibitor IL-18-binding protein a (IL-18BPa). The binding property of IL-18Bpa has a high affinity for IL-18, and it blocks the IL-18 interaction with the IL-18 receptor, inhibiting the IL-18 signaling pathway and the production of IFN-γ [16,17].

The ability of Inflammasome sensors, such as NLRP3, to mediate the secretion of IL-18, a cytokine that contributes to epithelial barrier repair against damage, is a mechanism that explains the protective role of IL-18 against colitis-associated colorectal cancer [17,18]. IL-18 may be associated with various pathologies, such as infections, allergic diseases and tumor immunity [16], but it also has protective functions in inflammatory diseases. Others have previously reported abundant IL-18 and caspase-1 plasmatic levels, as well as elevated mRNA levels for the same factors in the lesional skin of CTCL patients [19]. However, the expressions of other inflammasome components have not yet been evaluated.

Our findings demonstrated increased IL-18 in the skin dermal layer, lymph nodes and plasma levels of Sézary patients, contrary to decreased IL-1B in neoplastic lymph nodes and peripheral blood mononuclear cells (PBMCs). Taken together, we show, for the first time, a differential expression of IL-18 and IL-1B, along with the signaling-activation pathway lymph nodes, of advanced disease in Sézary patients.

## 2. Results

### 2.1. Expression of Inflammasome Components in the Skin of Patients with SS

The demographic data of the Sézary patients and control groups are included in Appendix A Appendix A. Herein, to verify whether the expression of inflammasome components could be altered by Sézary syndrome, we assessed the expression of NLRP1, NLRP3, AIM2, IL-1B and IL-18 by immunohistochemistry in the epidermal and dermal layers of skin from SS patients (Figure 1 and Appendix A Appendix A). Due to the usual erythroderma skin observed in SS patients, we included patients with idiopathic erythroderma (IE) in our cohort as a control group. The epidermal skin layers from SS patients showed an increased IL-1B expression and low levels of IL-18, as compared to the control groups (Figure 1B). However, in the dermal layer, an increased IL-18 expression was observed in SS individuals compared to the IE and HD groups, and similar IL-1B expression was observed among all groups (Figure 1C). Despite the abnormal expression of IL-1B and IL-18 observed in SS skin, NLRP3 and AIM-2 were equally expressed between the HD and SS samples (Appendix A Appendix A). Moreover, a decrease in NLRP1 expression was observed in SS skin compared to the HD group (Appendix A Appendix A). The findings showed an atypical expression of IL-1B and IL-18 that was verified in the skin of SS patients but not in the EI group, indicating that it may not be related to an erythroderma condition.

### 2.2. High Levels of Serum IL-18 and IL-18BPa in Sézary Syndrome

We assessed the serum levels of IL-18 and its inhibitor IL-18BPa in the subjects. The SS patients showed an increased production of IL-18 and IL-18BPa compared to healthy donors (Figure 2). However, the ratio of IL-18/IL18BPa was similar between the groups, which suggests that the levels of IL-18BPa could be due to the fact of negative feedback from IL-18, which may play a mitigating role. We previously verified that IL-1B serum levels were similar between the SS and HD groups [7].

### 2.3. Inflammasome Profile in PBMC from Sézary Patients

The findings indicated an aberrant IL-18 expression in the skin and at circulating levels in Sézary syndrome. Furthermore, the evidence for the increased expression of IL-18 with decreased IL-1B levels could represent distinct pathways of inflammasome activation, as well as a tumoral escape to avoid apoptosis. In Sézary syndrome, the neoplastic cells are present mainly in the blood and lymph nodes. Therefore, we analyzed inflammasome component expression in PBMCs from the SS and HD groups by RT-qPCR. The results revealed a decreased expression of *NLRP1*, *NLRP3* and *IL-1B* in PBMCs from SS patients and increased *NLRP4* transcripts levels compared to the HD samples (Figure 3A). Due to the fact that PBMCs include other cell types in addition to CD4+ T cells, we assessed the expression of inflammasome components in SS cell lines. The three SS cell lines evaluated (i.e., Hut78, SZ4 and SEAX) exhibited heterogeneous expression profiles of the transcripts (Figure 3B), including *CARD8*. The findings revealed altered expressions of some inflammasome components in PBMCs from SS patients.

### 2.4. Inflammasome Profile in Lymph Nodes of Sézary Patients

Sézary syndrome is characterized by significant blood involvement, erythroderma, and, often, lymphadenopathy. After verifying the expression of inflammasome components in the skin and the peripheral mononuclear blood cells, we next analyzed lymph-node biopsies from SS patients by immunohistochemistry (Figure 4 and Appendix A Appendix A).

Our analysis showed that lymph nodes from SS patients at the N2/N3 stages expressed lower levels of IL-1B compared to the N1 stage and IE nodes (Figure 4B). In contrast, enhanced IL-18 expression was observed in N2/N3 lymph nodes in SS patients compared to the IE group (Figure 4B). In addition, NLRP1, NLRP3 and AIM2 were equally expressed at the protein level among the groups (Appendix A Appendix A). We detected decreased IL-1B levels, which were associated with the increased protein levels of IL-18 in the different components analyzed, including skin, PBMCs and the lymph nodes.

Moreover, we analyzed the whole transcript profile of lymph nodes from both groups (i.e., SS and IE) by RNA sequencing. Appendix A Appendix A illustrates the unsupervised z-score heatmap of the normalized counts of the 300 most variable and expressed genes. The samples were segregated according to their respective groups and presented a similar expression profile. We also evaluated the expression of 58 genes involved in the inflammasome function (Figure 5), in addition to *IL1B* and *IL18*. Our transcriptome analysis showed low RNA levels of *IL1B* in SS lymph nodes, which was in line with the protein-level observations. However, in addition to the atypical IL-18 protein expression in the lymph nodes of SS patients, the same *IL18* RNA levels were found between the groups. Among the 58 genes involved in the inflammasome function, we identified 26 genes with a fold-change (FC) variance of up to 1.3 (i.e., 17 genes showed FC ≤ −1.3 and 9 genes FC ≥ 1.3) compared to the IE group, including the key players in initiating the inflammasome activation, such as *CD40LG*, *NLRP3* and *NLRC5*, which were all downregulated DEGs. The *NLRP1* and *AIM2* genes, which also initiate the formation of an inflammasome, in addition to *NLRP3*, were equally expressed between the SS and IE nodes, which was in line with the protein-level observations.

As our immunohistochemistry assays confirmed the aberrant protein expression of IL-1B and IL-18 in the SS lymph nodes, we speculated whether the corresponding signaling pathways would also be affected. Therefore, we built a gene network for the IL-18 and IL-1B signaling pathways (Appendix A Appendix A). In addition to the downregulation of *IL1B* itself, the diverse genes related to *IL1B* were also downregulated, such as the anti-inflammatory antagonist of IL-1 proinflammatory cytokines (*IL1RN*) and the components related to NFκB pathway activation, such as *NFKB2*, *FOS*, *FOSB*, *JUN*, *JUNB* and *IRAK2* (including the NFκB inhibitors, *NFKBIZ* and *NFKB1A*). Furthermore, the AP-1 transcription factors *FOS*, *FOSB*, *JUN* and *JUNB*, together with *EGR1*, configured the top 10 most variable genes of the IL-1B and IL-18 signaling network genes. The other genes related to NFκB pathway activation, such as *Myd88*, *NFKB1* and *RELA*, exhibited FC ≤ −1.3 but without a significant *p*-value. It was interesting that the IL-1B receptors, *IL1RL1* and *IL1R1*, were among the upregulated genes in the signaling network.

The present findings, summarized in Figure 6, showed a compartmentalized expression of IL-1B and IL-18 and provide the first evidence of an imbalanced expression of IL-1B and IL-18, as well as their associated regulators of inflammation, in Sézary syndrome.

## 3. Discussion

We assessed the expression of inflammasome components at the protein and transcript levels at sites of tumoral burden in SS patients, including skin, peripheral blood and lymph nodes. We observed atypical IL-1B and IL-18 protein expressions in the epidermal and dermal skin layers of SS patients. In the epidermal layer, we noticed an enhanced IL-1B protein expression. IL-1B has an important proinflammatory role in the immune response activated by bacteria, and to this extent, the upregulation of IL-1B observed at the SS epidermal layer could be due to the fact of continuous activation by bacterial antigens. Furthermore, skin infections and large colonizations by *S. aureus* are frequent in SS [20]. In the dermal layer, IL-1B expression was similar between the groups, but IL-18 was increased compared to the IE group. Erythroderma is a manifestation of a wide range of cutaneous and systemic diseases, including infection, malignancy and drug hypersensitivity [21]. Unfortunately, it was not possible to evaluate the inflammasome components NLRP1, NLPR3 and AIM-2 in IE skin samples. However, the cases of idiopathic erythroderma skin samples did not show changes in the IL-1B and IL-18 protein expressions in the epidermis or dermis layer when compared to healthy donors, which suggests no inflammasome activation despite the erythroderma condition. Nevertheless, others inflammatory mechanisms, such as caspase-8 triggered by fas-ligand signaling [22], proIL-18 conversion by chymases secreted by mast cells [23] or by granzyme B derived from CD8+ T cells [23,24], can also lead to maturation and secretion of IL-18. Therefore, other inflammatory mechanisms could be involved in the inflammatory process, so additional investigations are needed to understand its relationship with the condition.

The downregulation of NLRP1 at the protein level was also noticed in the epidermal layer of SS patients, as compared to the HD controls. NLRP1 is considered the principal inflammasome sensor in human keratinocytes [25], and it is activated upon UVB radiation exposure [26]. Phototherapy, including UVB radiation exposure, has been broadly used as a monotherapy for the early stages of CTCL and has also been combined with other systemic therapies for CTCL patients with advanced disease, such as those with SS [27]. It is known that human keratinocytes respond to UVB exposure by secreting large amounts of IL-1B [28]. However, more studies are necessary to clarify the contribution of NLRP1 activation in the clearance of CTCL.

Lymph-node biopsies in SS patients with suspicion of lymph node involvement were also evaluated regarding inflammasome component expression at the protein and transcript levels. We evaluated N1 stage nodes with dermatopathic lymphadenopathy; N2 stage nodes with the presence of neoplastic cells without the loss of the lymph node architecture; and N3 stage nodes with alteration of the lymph node architecture, where these stages (N2/N3) characterize the advanced stages of the disease. The neoplastic involvement of lymph nodes in SS patients was accompanied by a decrease in IL-1B, which was associated with the upregulation of IL-18 protein expression compared to IE. Furthermore, the gene expression differential analysis of the lymph nodes was consistent with the IL-1B downregulation, along with other components of the NFκB signaling pathway, such as *NFKB2* and *IRAK2* [29]. The NFκB pathway plays an important role in bacterial infection resolution [30], which is common in individuals with SS [31]. A downregulation of this pathway could directly impair the inflammatory response against infections in these patients.

The expressions of inflammasome components were also atypical in the PBMCs of SS patients. Even though the PBMCs’ composition included monocytes and dendritic cells, in addition to the lymphocytes, the SS patients had increased levels of CD4 T cells and increased CD4/CD8 ratios of ≥10, which indicated the high presence of neoplastic cells in the peripheral blood. Our results showed the downregulation of the *NLRP1* transcript on PMBCs from SS patients, which was in agreement with our protein-level observations in the skin of SS patients.

Our results also showed an overexpression of *CARD8* in the PBMCs of SS patients, as compared to the IE samples. *CARD8* was also expressed in the SS cell lines analyzed, showing a similar profile to PBMCs of SS patients. The activation of CARD8 inflammasome may trigger pyroptosis in human T cells under healthy conditions. Moreover, a CARD8-induced pyroptosis pathway can only be engaged in quiescent T cells and, thus, not in activated T cells [32,33]. There is no evidence regarding the role of CARD8 in SS cells, so additional investigations are needed to understand its relationship with the disease.

It is known that the activity of IL-18 is balanced by the presence of IL-18BPa, and serum IL-18BPa levels have been significantly elevated in sepsis and other inflammatory diseases [34,35,36]. Herein, we detected high serum levels of IL-18 in SS patients, along with an increased expression of protein in the dermal skin layer and in the N2/N3 nodes. Overall, high levels of IL-18BPa were detected. Moreover, we had previously observed that IL-1B was scarcely detected in the serum of SS patients [7].

While we observed an upregulation of IL-18, the increased levels of its regulator, IL-18BPa, could inhibit IL-18 proinflammatory functioning in SS patients. It was previously described that CTCL patients showed elevated plasma levels of IL-18 and caspase-1, regardless of the disease stage, and IL-18 and caspase-1 were overexpressed in lesioned skin, at the protein and mRNA levels [19]. High levels of IL-18 in the absence of IL-12 could contribute to the Th2 skewing evident in this disease [19]. We suspect that the inhibition of IL-1B could be a tumoral evasion of pyroptosis.

The inflammasome signaling and cell secretion of mature IL-1B are often accompanied by pyroptosis. The danger detection and immune alert system may lead to inflammasome hyperactivation. Their counter-regulatory mechanism is crucial to avoid IL-1B, since it is a key mediator of the inflammatory response. Furthermore, we observed a reduction in the transcriptional and protein expressions of IL-1B in SS patients, which may suggest an attempt to evade hyperactivation by inflammasomes. The resistance to apoptosis is a fundamental feature of malignant cells, and several mechanisms are described to induce apoptosis resistance in CTCL, such as the activation of the NF-κB pathway [36], delayed expression of FasL following activation [37], diminished FAS/CD95 expressions [38], downregulation of the TRAIL pathway [39] and loss of TNFR1 with a high level of IER3 expression, among other factors [38].

Human primary T cells are fully competent for conventional inflammasome signaling, which governs the execution of pyroptosis [33]. The candidate components in primary T cells are dependent on the CARD8–caspase-1–GSDMD axis, and T cells committed to malignant transformation could benefit from shutting down this programmed cell death pathway. Whether the downregulation of IL-1B could be a tumoral escape to avoid pyroptosis provides a potential avenue to better understand the immunopathogenesis of Sézary syndrome.

## 4. Materials and Methods

### 4.1. Human Subjects

This study included SS patients (*n* = 28; 16 M/12 F; median age of 61 years, ranging from 28 to 84 years;) diagnosed with SS between 2014 and 2022, followed up in the Cutaneous Lymphomas Clinic of the Hospital das Clínicas, Division of Clinical Dermatology, University of São Paulo Medical School, in Brazil (HCFMUSP). All SS patients exhibited de novo erythroderma, abd none of the cases were derived from mycosis fungoides progression. All patients fulfilled the criteria proposed by the International Society for Cutaneous Lymphomas (ISCL) and the cutaneous lymphoma task force of the European Organization of Research and Treatment of Cancer (EORTC) for the diagnosis of SS: the same monoclonal T-cell population detected on blood and skin (mandatory criterion) plus CD4/CD8 ≥ 10 and/or CD4+ CD7− ≥ 40%, CD4+ CD26− ≥ 30%, or ≥ 1000 Sézary cells/µL detected on peripheral blood smear [40]. Erythrodermic patients who did not fulfill the criteria proposed by the ISLC/EORTC were excluded from the study.

Peripheral blood (*n* = 21; 11 M/10 F) and paraffin skin biopsies (*n* = 15; 7 M/8 F) were collected from the SS patients. Lymph node samples (*n* = 8; 4 M/4 F; N1 stage = 4, N2/N3 stage = 4) were also obtained from excisional biopsy after evaluation of the neoplastic involvement of the lymph nodes by clinical examination, computed tomography or positron emission tomography. The criteria established for suspicion of the neoplastic involvement of the lymph nodes were firm consistency, irregular shape, clustered presentation, fixed position or a diameter ≥ 1.5 cm.

Idiopathic erythroderma patients were established as the erythrodermic control group (IE group, *n* = 19; 13 M/6 F; median age of 65 years, ranging from 32–77 years). Paraffin skin biopsies were collected from IE patients (*n* = 12; 10 M/2 F), and lymph nodes samples were obtained from the IE patients with dermatopathic lymphadenopathy (*n* = 8; 4 M/4 F).

Healthy donor (HD group, *n =* 40, 18 M/22 F; median age of 54 years, ranging from 30 to 85 years) samples were used as the control group for skin and peripheral blood analysis. Peripheral blood (*n =* 28, 14 M/14 F) and paraffin skin biopsies (*n* = 12; 4 M/8 F) were obtained from the HDs.

The study inclusion criteria required that patients be over 18 years old with established Sézary syndrome, and samples were obtained before treatment commenced. Patients with dermatologic diseases, those currently being treated and those with a history of autoimmune diseases were not included in this evaluation.

### 4.2. PBMC and Cell Lines

PBMCs were isolated from heparinized venous blood by Ficoll–Hypaque gradient centrifugation (GE Healthcare, Uppsala, Sweden) and frozen in RNAlater ™ solution (Invitrogen, Waltham, MA, USA) for further analysis.

All cells were cultured in an RPMI-1640 medium (Sigma, St. Louis, MO, USA) with 1% penicillin/streptomycin (Sigma) at 37 °C with 5% CO_2_. The cell lines used in this study have been described elsewhere [41,42,43]. Hut78 was supplemented with 10% fetal bovine serum (FBS) (Sigma); meanwhile, SeAx and SeZ4 were supplemented with 10% AB human serum (Sigma) and 5 ng/mL of human recombinant IL-2 and IL-4 (Thermo Fisher Scientific, Waltham, MA, USA). The cell lines were a generous gift from Professor N. Ødum (Copenhagen, Denmark).

### 4.3. RNA Extraction, Sequencing and Transcriptomic Analyses

The extraction of total RNA was performed with the RNeasy Plus Mini Kit (Qiagen, Hilden, Germany), according to the manufacturer’s recommendations. The sample quality was verified by TapeStation (Agilent, Santa Clara, CA, USA). The gene expression was analyzed by RNA sequencing using Illumina TruSeq Stranded (San Diego, CA, USA). The adaptors were removed, and the sequence reads were submitted to the quality assurance program FastQC [44]. The resulting sequences were mapped against the reference human genome assembly (GRCh38.p13 release 107) with the program Subread [45], following counting with featureCounts [46]. The quality assessment, normalization, statistical analysis and identification of differentially expressed genes (DEGs) were performed using the R/Bioconductor DESeq2 package [47]. The *p*-values were evaluated for false discovery rate adjustment using the Benjamini–Hochberg method. Genes were considered differentially expressed if *p*-value ≤ 0.05. Over-representation analysis was performed using the program Enrichr, and the gene network was built with Cytoscape, using the InnateDB protein–protein interaction database [48,49,50].

### 4.4. Gene Expression by Real-Time Polymerase Chain Reaction

Total RNA was measured using a NanoDrop ND-1000 spectrophotometer (Thermo Fisher Scientific), and a reverse-transcription reaction was performed with the iScriptTM kit (Bio-Rad, Hercules, CA, USA). For real-time polymerase chain reaction, complementary DNA was incubated with SYBR Green (Applied Biosystem, Waltham, MA, USA) and the primers for all target genes. All primer sequences for NLRP1, NLRP3, NLRP4, IL-18, IL-1B and CARD8 were synthesized by Invitrogen, and the sequences are listed in Appendix A Appendix A. The DNA amplification was carried out in a 7500 real-time PCR system (Applied Biosystems, Waltham, MA, USA), and data analysis was performed using the 7500 software, version 2.0.6 (Applied Biosystems), according to the ΔΔ-cycle threshold method [51].

### 4.5. Immunohistochemistry

Each skin sample was formalin-fixed and then embedded in paraffin and sectioned at 4 μm. The histological sections were deparaffinized in xylol, rehydrated in ethanol and blocked with 3% hydrogen peroxide. The tissue sections were then incubated with the primary antibodies polyclonal anti-NLRP1 rabbit antibody (Ab 3683); monoclonal mice anti-NLRP3 (214185); polyclonal anti-AIM2 rabbit antibody (93015); polyclonal anti-IL-18 rabbit antibody (Ab71495); and polyclonal anti-IL-1B rabbit antibody (Ab2105), all of which were from Abcam (Cambridge, MA, USA). The Reveal Biotin-Free Polyvalent HRP system (SPB-999, Spring Bioscience Corp, Pleasanton, CA, USA) was selected as the amplification system, and DAB (3,3’ diamibenzidine, D5637, Sigma) was used for the revelation. Negative reaction controls were obtained by omitting the primary antibody and replacing it with PBS pH 7.4. After mounting the slides with Permount resin (FISHER Scientific, Fair Lawn, NJ/USA), the slides were scanned using an Aperio ScanScope slide scanner (Aperio Technologies, Vista, CA, USA). Then, the images were analyzed with Image-Pro Plus, version 4.5.0.29 (Media Cybernetics Inc., Bethesda, MD, USA).

### 4.6. Determination of IL-18 and IL-18BPa

The serum IL-18 and IL-18BPa levels were evaluated by an enzyme-linked immunosorbent assay (ELISA) (R&D System, Minneapolis, MN, USA) with a detection limit of 11.7 and 93.8 pg/mL, respectively.

### 4.7. Statistical Analysis

Comparisons between the two groups were performed according to the Mann–Whitney test. The level of significance considered was *p* ≤ 0.05. All of the statistical and graphic representations were executed with GraphPad Prism 7 software (Graphpad Holdings, San Diego, CA, USA).

## Figures and Tables

**Figure 1 ijms-24-04674-f001:**
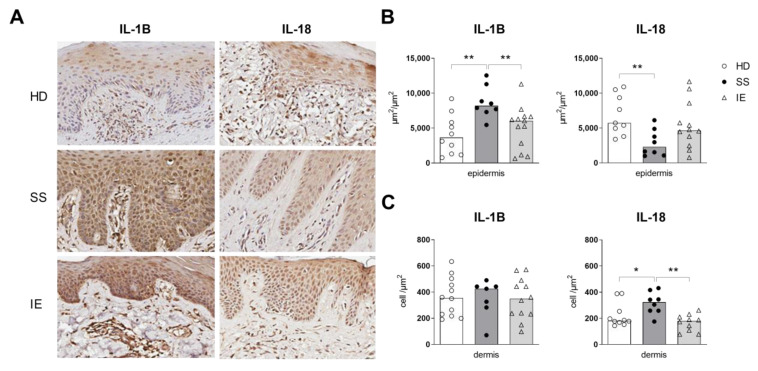
Increased protein levels of epidermal IL-1B and dermal IL-18 in skin of erythrodermic SS. The expression of IL-1B and IL-18 (**A**) were assessed in the skin biopsies of SS patients (*n* = 7–8, closed circle), healthy donors (HDs, *n* = 9–12, open circle), and idiopathic erythroderma (IE, *n* = 9–13, open triangle) by immunohistochemistry. The analyses were performed on the epidermis (**B**) and dermis (**C**). The values are expressed as medians. * *p* ≤ 0.05 and ** *p* ≤ 0.01.

**Figure 2 ijms-24-04674-f002:**
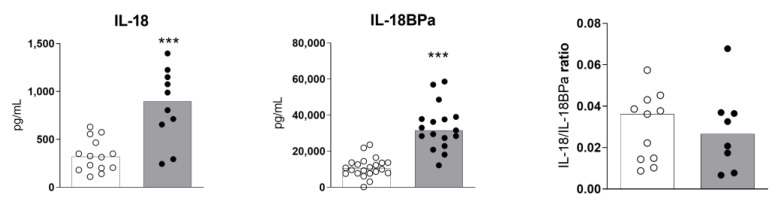
High circulating levels of IL-18- and IL-18-binding protein a (BPa) in SS patients. Serum determinations of IL-18 and IL-18BPa were assessed by ELISA (SS, *n* = 10–19; HD, *n* = 13–22). The IL18/IL18BP ratios were also calculated. The values are expressed as medians. *** *p* ≤ 0.001.

**Figure 3 ijms-24-04674-f003:**
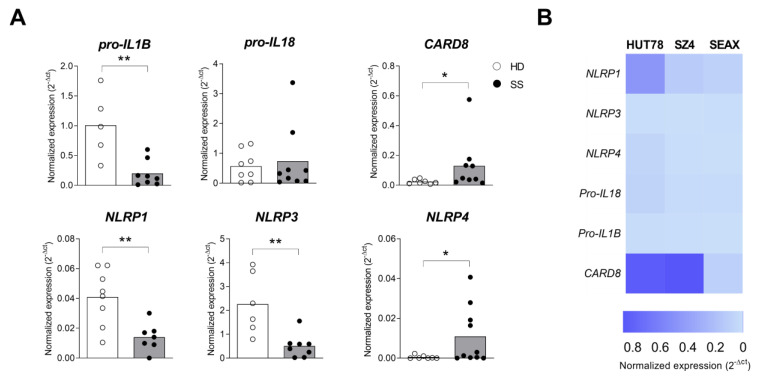
Altered inflammasome transcript expressions in peripheral blood mononuclear cells from SS patients. Expressions of *NLRP1*, *NLRP3*, *NLRP4*, *pro-IL18*, *pro-IL1B* and *CARD8* were analyzed in PBMCs from SS patients (*n* = 7–10, closed circle), healthy control individuals (*n* = 5–8, open circle) and in SS cell lines by qRT-PCR. The data were normalized to *GAPDH* and shown either by median (**A**) for SS patient samples or by heatmap (**B**) in SS cell lines. * *p* ≤ 0.05 and ** *p* ≤ 0.01.

**Figure 4 ijms-24-04674-f004:**
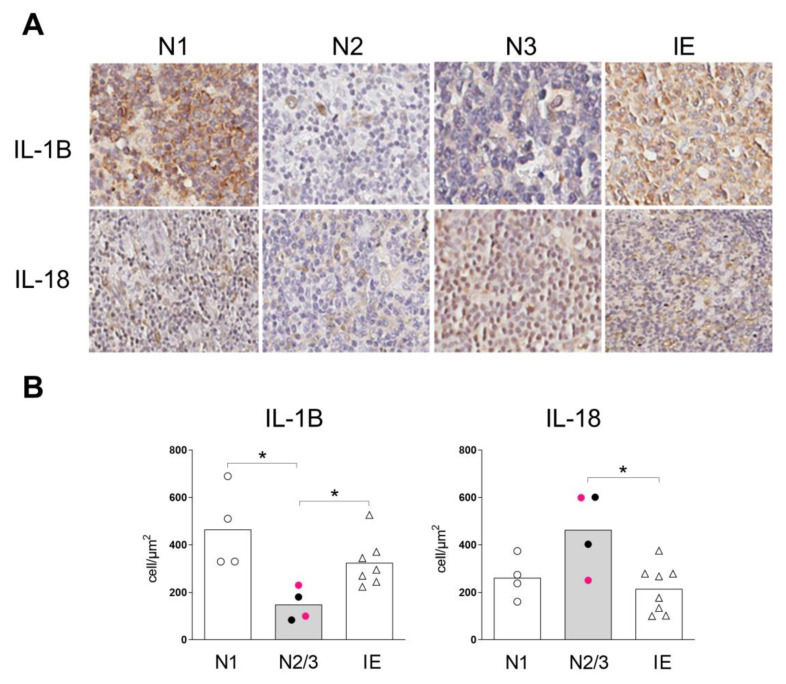
Decreased IL-1B expression and IL-18 upregulation in lymph nodes at an advanced stage of SS. The expressions of IL-1B and IL-18 were assessed in the lymph-node biopsies of SS patients at stages N1 (*n* = 4, open circle), N2 (*n* = 2, pink circle), and N3 (*n* = 2, closed circle), as well as in idiopathic erythroderma patients (IE, *n* = 8, open triangle), by immunohistochemistry. (**A**) Representative images, original magnification ×20. (**B**) IL-1B expression and IL-18 expression. Data are illustrated by median values. * *p* ≤ 0.05.

**Figure 5 ijms-24-04674-f005:**
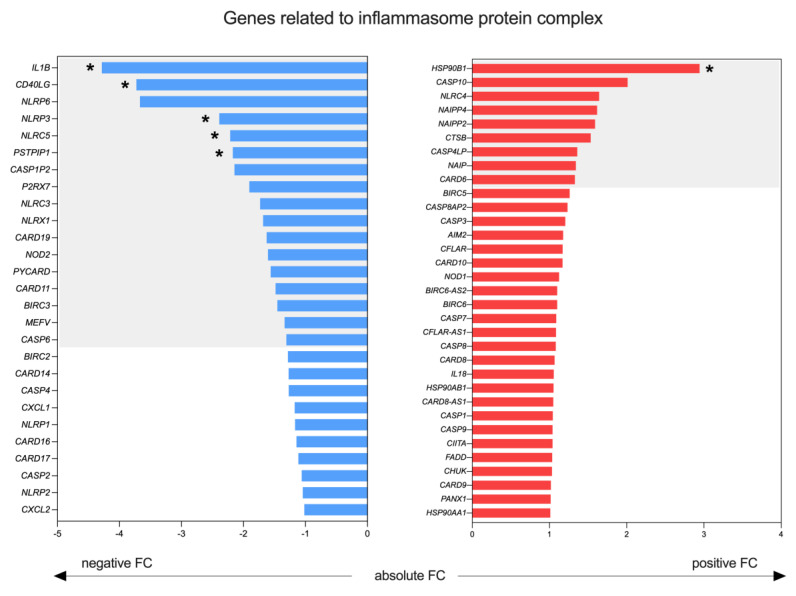
Expression of key genes involved in inflammasome function in the lymph nodes of SS patients. The expression profiles of SS lymph nodes, as compared to IE lymph nodes, were evaluated by RNA sequencing. The absolute FC values of 60 key genes involved in inflammasome function are shown in blue (FC < −1.00) or red (FC > 1.00) bars. The colored background areas represent genes that have either FC ≥ 1.3 or FC ≤ −1.3. * Indicates a *p*-value significance level of 0.05.

**Figure 6 ijms-24-04674-f006:**
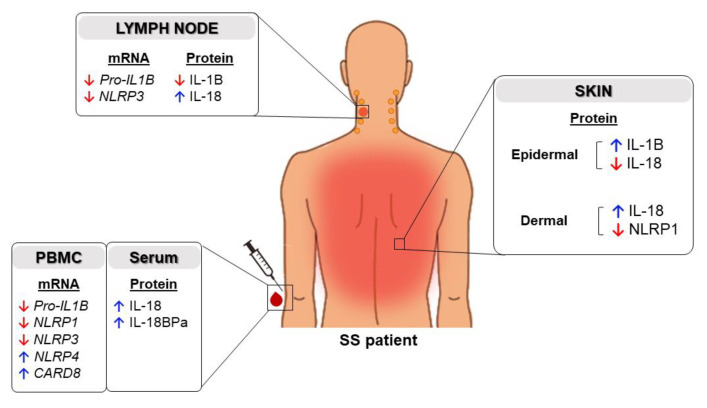
Imbalanced expression of IL-1B and IL-18, as well as their associated regulators of inflammation, in Sézary syndrome.

## Data Availability

Data is unavailable due to privacy restrictions.

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
