# Peer review of "Imbalanced IL-1B and IL-18 Expression in Sézary Syndrome"

_ijms, 2023, doi:10.3390/ijms24054674_

Round 1

Reviewer 1 Report

The Inflammasome-mediated changes in IL-1β and IL-18 have been reported in immunopathogenesis. The two cytokines are activated by the inflammasomes. In this study, authors report a differential expression of IL-1β and IL-18 in skin and lymph nodes of Sezary syndrome (SS) patients. The study reports an increase in epidermis IL-1β (while IL-18 decreased), but, as indicated by the transcriptomic analysis, IL-1β decreased in idiopathic erythroderma nodes of SS patients, where a decrease in expression of NLRP3 was also reported.

At the outset, it is an interesting study on a rare lymphoma. I have certain observations/ remarks:

(1)   The Abstract reads, “Our findings at the skin show increased IL-1β and decreased IL-18 protein expression in the epidermis of SS patients. An inverse picture is seen in the lymph nodes, with down-regulation of IL-1β and an enhancement of IL-18 at protein level in nodes at the advanced stage of SS”. On the other hand, the last paragraph of the Introduction reads, “Our findings demonstrated an increased IL-18 in skin dermal layer, lymph nodes and plasma levels of Sézary patients”. Does it mean the IL-18 increased in both dermal layer and lymph node?

(2)   The length of Introduction may be cut short, more focused on IL-1β and IL-18.

(3)   Authors may like to use more commonly used term in place of ‘Casuistic’. May clearly define the inclusion and exclusion criteria. How the no. n=28 was arrived at?

(4)   Page 7, Line 1: The full form of PBMC should be give where the abbreviation was first used. Authors first used the abbreviation in Introduction section.

(5)   Authors should ideally give statement about tissue/biopsy fixation method for immunohistochemistry.

(6)   In discussion section, authors need to elaborate their findings on inflammasome expression in idiopathic erythroderma skin samples. What other inflammatory mechanisms may work?

Author Response

Comments and Suggestions for Authors

The Inflammasome-mediated changes in IL-1β and IL-18 have been reported in immunopathogenesis. The two cytokines are activated by the inflammasomes. In this study, authors report a differential expression of IL-1β and IL-18 in skin and lymph nodes of Sezary syndrome (SS) patients. The study reports an increase in epidermis IL-1β (while IL-18 decreased), but, as indicated by the transcriptomic analysis, IL-1β decreased in idiopathic erythroderma nodes of SS patients, where a decrease in expression of NLRP3 was also reported.

At the outset, it is an interesting study on a rare lymphoma. I have certain observations/ remarks:

  • The Abstract reads, “Our findings at the skin show increased IL-1β and decreased IL-18 protein expression in the epidermis of SS patients. An inverse picture is seen in the lymph nodes, with down-regulation of IL-1β and an enhancement of IL-18 at protein level in nodes at the advanced stage of SS”. On the other hand, the last paragraph of the Introduction reads, “Our findings demonstrated an increased IL-18 in skin dermal layer, lymph nodes and plasma levels of Sézary patients”. Does it mean the IL-18 increased in both dermal layer and lymph node?

Thank you for the comments. Yes, it means that IL-18 expression is increased in both dermal layer and lymph node. We include this information in the abstract.

  • The length of Introduction may be cut short, more focused on IL-1β and IL-18.

We have shortened the introduction (focusing on IL-1β and IL-18).

  • Authors may like to use more commonly used term in place of ‘Casuistic’. May clearly define the inclusion and exclusion criteria. How the no. n=28 was arrived at?

The word “casuistic” was changed for “human subjects”. The whole section “human subjects” was better described, including the inclusion and exclusion criteria (bellow).

“This study included SS patients (n = 28; 16M/12F; median age of 61 years, ranging from 28-84 years;) diagnosed with SS between 2014 and 2022, followed up in the Cutaneous Lymphomas Clinic of the Hospital das Clínicas, Division of Clinical Dermatology, University of São Paulo Medical School in Brazil (HCFMUSP). All SS patients exhibited de novo erythroderma, none of the cases were derived from mycosis fungoides progression. All patients fulfilled the criteria proposed by the International Society for Cutaneous Lymphomas (ISCL) and the cutaneous lymphoma task force of the European Organization of Research and Treatment of Cancer (EORTC) for the diagnosis of SS: the same monoclonal T-cell population detected on blood and skin (mandatory criterion) plus CD4/CD8 ≥ 10 and/or CD4+CD7- ≥ 40%, CD4+CD26- ≥ 30%, or ≥ 1000 Sézary cells/µL detected on peripheral blood smear”.

The samples distribution over the assays was also included in the Supplementary Table 2 - Subjects characteristics.

  • Page 7, Line 1: The full form of PBMC should be give where the abbreviation was first used. Authors first used the abbreviation in Introduction section.

It was corrected.

  • Authors should ideally give statement about tissue/biopsy fixation method for immunohistochemistry.

We included as follow:  Each skin sample was formalin-fixed and then embedded in paraffin and sectioned at 4 μm

  • In discussion section, authors need to elaborate their findings on inflammasome expression in idiopathic erythroderma skin samples. What other inflammatory mechanisms may work?

The skin samples of idiopathic erythroderma patients (IE group) showed similar IL-1b and IL-18 protein expression when compared to skin samples of health donors (HD group). As the cytokines IL-1b and IL-18 are considered potential markers of inflammasome activation we hypothesize that idiophathic erytroderma may not related to inflammasome activation. Unfortunately, it wasn’t possible to evaluate inflammasome components (NLRP1, NLPR3 and AIM-2) in IE skin samples to provide a more complete picture of inflammasome expression in IE samples. Nevertheless, others inflammatory mechanisms, such as caspase-8 triggered by fas-ligand signaling, proIL-18 conversion by chymases secreted by mast cells or by granzyme B derived from CD8+ T cells, can also lead to maturation and secretion of IL-18. Therefore, other inflammatory mechanisms could be involved in the inflammatory process and further investigation is needed to address this question.  

The topic was included in the discussion.

Reviewer 2 Report

The authors assessed the skin, serum, peripheral mononuclear blood cells (PBMC) and lymph node samples of SS patients and control groups, to investigate expression of IL-1β and IL-18 at protein and transcript expression levels, as a redout of inflammasome activation. The observed increased IL-1β and decreased IL-18 protein expression in the epidermis of SS patients but an inverse picture is seen in the lymph nodes. Of notice, transcriptomic analysis on SS and IE nodes confirms a decreased expression of IL-1β and NLRP3, whereas pathway analysis further shows downregulation of IL-1β-associated genesTheir findings show a compartmentalized expression of IL-1β and IL-18 and provide first evidence of an imbalanced expression of IL-1β and IL-18 in Sezary syndrome patients.

The first subheading in the materials and methods says "casuistic". While that is an English word, I do not think it is the one you want for the subheading as the meaning of this word in English does not seem appropriate for the content of that section.

In Figure 4 the last row of histological images has "IE" while the graphs below have "EI".

It looks like the authors have made significant changes to the document from past submissions. I am not sure why so much data is provided as supplemental information in this version.

Supplemental figure 4 is very complex and unlikely to provide significant information to the average reader other than to show the complexity.

While many changes were made, the flow of the information presented in the paper could still be improved. The introduction reads a bit like a collection of points which are not always properly linked together. The same occurs with the presentation of the data where a connection between the different data presented was difficult on an initial read. However, the conclusion is simple and the data supports it.

Author Response

Comments and Suggestions for Authors

The authors assessed the skin, serum, peripheral mononuclear blood cells (PBMC) and lymph node samples of SS patients and control groups, to investigate expression of IL-1β and IL-18 at protein and transcript expression levels, as a redout of inflammasome activation. The observed increased IL-1β and decreased IL-18 protein expression in the epidermis of SS patients but an inverse picture is seen in the lymph nodes. Of notice, transcriptomic analysis on SS and IE nodes confirms a decreased expression of IL-1β and NLRP3, whereas pathway analysis further shows downregulation of IL-1β-associated genes. Their findings show a compartmentalized expression of IL-1β and IL-18 and provide first evidence of an imbalanced expression of IL-1β and IL-18 in Sezary syndrome patients.

  • The first subheading in the materials and methods says "casuistic". While that is an English word, I do not think it is the one you want for the subheading as the meaning of this word in English does not seem appropriate for the content of that section.

Thank you for the comments. The word “casuistic” was changed for “human subjects”.

  • In Figure 4 the last row of histological images has "IE" while the graphs below have "EI".

It was corrected.

  • It looks like the authors have made significant changes to the document from past submissions. I am not sure why so much data is provided as supplemental information in this version.

A considerable part of the manuscript was reformulated between the first and second submissions, including some data analysis. In this process, we decided to keep data that was not essentially crucial to the plot, but still relevant in the big picture, as supplementary material (Supplementary Figures 1 and 2). Supplementary Table 1 presents technical information, while Supplementary Table 2 and Supplementary Figure 3 present subject characteristics and general information about the transcriptomic data set. And finally, Supplementary figure 4 brings additional information to the data set.

  • Supplemental figure 4 is very complex and unlikely to provide significant information to the average reader other than to show the complexity.

We agree with the reviewer that Supplementary Figure 4 is too complex and difficult to extract information due to the amount of data. However, Supplementary Figure 4 also illustrates the predominance of the blue color (down-regulated genes in the SS group) in the IL-1b and IL-18 network, especially IL-1b related genes. We consider this finding relevant and as it was commented several times in the manuscript, we preferred not to remove the figure even though it is difficult to read at first sight.

  • While many changes were made, the flow of the information presented in the paper could still be improved. The introduction reads a bit like a collection of points which are not always properly linked together. The same occurs with the presentation of the data where a connection between the different data presented was difficult on an initial read. However, the conclusion is simple and the data supports it.

We appreciate the comments. The manuscript has been extensively proofread in English, which we hope will make it easier to read. We also worked on the links in this version.

Round 2

Reviewer 1 Report

Revised version may be accepted.

Reviewer 2 Report

Thank you for the corrections and improvements on the text